# Intestinal Obstruction as Initial Presentation of Idiopathic Portal and Mesenteric Venous Thrombosis: Diagnosis, Management, and Literature Review

**DOI:** 10.3390/diagnostics14030304

**Published:** 2024-01-30

**Authors:** Bogdan Stancu, Alexandra Chira, Horațiu F. Coman, Florin V. Mihaileanu, Razvan Ciocan, Claudia D. Gherman, Octavian A. Andercou

**Affiliations:** 12nd Department of General Surgery, University of Medicine and Pharmacy “Iuliu Hațieganu”, 400006 Cluj-Napoca, Romania; bstancu7@yahoo.com (B.S.); ms26rfl@yahoo.com (F.V.M.); andercou@yahoo.com (O.A.A.); 2Department of Internal Medicine, 2nd Medical Clinic, University of Medicine and Pharmacy “Iuliu Hațieganu”, 400006 Cluj-Napoca, Romania; 3Department of Vascular Surgery, County Clinical Emergency Hospital, 400006 Cluj-Napoca, Romania; horatiucoman@yahoo.com; 4Department of Surgery—Practical Abilities, University of Medicine and Pharmacy “Iuliu Hațieganu”, 400337 Cluj-Napoca, Romania; razvan.ciocan@yahoo.com (R.C.); ghermanclaudia@yahoo.com (C.D.G.)

**Keywords:** thrombosis, mesenteric vein, portal vein, ischemia, intestinal obstruction, intestinal ileus

## Abstract

It is quite common for portal vein thrombosis to occur in subjects who present predisposing conditions such as cirrhosis, hepatobiliary malignancies, infectious or inflammatory abdominal diseases, or hematologic disorders. The incidence of idiopathic portal vein thrombosis in non-cirrhotic patients remains low, and despite the intensive workup that is performed in these cases, in up to 25% of cases, there is no identifiable cause. If portal vein thrombosis is untreated, complications arise and include portal hypertension, cavernous transformation of the portal vein, gastroesophageal and even small intestinal varices, septic thrombosis, or intestinal ischemia. However, intestinal ischemia develops as a consequence of arterial thrombosis or embolism, and the thrombosis of the mesenteric vein accounts for about 10% of cases of intestinal ischemia. Although acute superior mesenteric vein thrombosis can cause acute intestinal ischemia, its chronic form is less likely to cause acute intestinal ischemia, considering the possibility of developing collateral drainage. Ileus due to mesenteric venous thrombosis is rare, and only a small number of cases have been reported to date. Most patients experience a distinct episode of acute abdominal pain due to ischemia, and in the second phase, they develop an obstruction/ileus. Acute superior mesenteric venous thrombosis is a rare condition that is still associated with a high mortality rate. The management of such cases of superior mesenteric venous thrombosis is clinically challenging due to their insidious onset and rapid development. A prompt and accurate diagnosis followed by a timely surgical treatment is important to save patient lives, improve the patient survival rate, and conserve as much of the patient’s bowel as possible, thus leading to fewer sequelae.

## 1. Introduction

Acute mesenteric ischemia (AMI) is defined as an abrupt cessation of the mesenteric blood supply to the intestine with the development of symptoms that may vary with the time of onset, from minutes (in embolism) to hours (in atherothrombosis), leading to cellular damage, intestinal necrosis, and commonly patient death, if untreated.

There are four different etiological categories of AMI that need to be distinguished as they differ in treatment and prognosis. These are superior mesenteric artery (SMA) embolism (50%), atherosclerotic SMA occlusion (thrombosis) (15–25%), non-occlusive mesenteric ischemia (NOMI) (20%), and mesenteric venous thrombosis (5–15%). SMA embolism and thrombosis are often referred to as an arterial AMI or an occlusive AMI. The overall incidence is low (0.09–0.2% of all acute admissions to emergency departments), representing an infrequent cause of abdominal pain but a common cause of emergent intestinal resection. Prompt diagnosis and intervention are essential to reduce the mortality rates, which exceed 50% [1].

Acute mesenteric emboli can originate from the left atrium (e.g., atrial fibrillation), left ventricle (e.g., left ventricular dysfunction with a poor ejection fraction), or cardiac valves (e.g., endocarditis). Occasionally, emboli are generated from an atherosclerotic aorta. Emboli typically lodge at points of normal anatomic artery narrowing, usually at 3–10 cm distal to the origin of the SMA, thus sparing the proximal jejunum and colon. More than 20% of SMA emboli are associated with emboli concurrent with other arteries including those of the spleen and the kidneys [2].

The thrombosis of the SMA is usually associated with a pre-existing chronic atherosclerotic disease leading to stenosis. Many of these patients have a history consistent with chronic mesenteric ischemia (CMI), including postprandial pain, weight loss, or “food fear”. A detailed medical history is important when evaluating a patient suspected of having AMI. Thrombosis usually occurs at the origin of visceral arteries. An underlying plaque in the SMA usually progresses eventually to a critical stenosis, resulting in collateral beds. Accordingly, symptomatic SMA thrombosis most often accompanies celiac occlusion. SMA thrombosis may also occur due to vasculitis, mesenteric dissection, or mycotic aneurysm. The involvement of the ileocolic artery will result in the necrosis of the proximal colon [3].

Acute non-occlusive mesenteric ischemia (NOMI) is usually a consequence of SMA vasoconstriction associated with a low splanchnic blood flow. The compromised SMA blood flow also affects the proximal colon due to the involvement of the ileocolic artery. Patients with NOMI typically suffer from severe coexisting illness, commonly cardiac failure, which may be precipitated by sepsis. Hypovolemia and the use of vasoconstrictive agents may precipitate NOMI [4].

Mesenteric venous thrombosis (MVT) is an uncommon cause of intestinal ischemia. Its usual presentation is acute abdominal pain that is out of proportion compared to other abdominal symptoms. In its severe form, it may present clinical symptoms suggestive of perforation and peritonitis. A new onset of ascites may also raise a suspicion of MVT. In rare cases where ischemia continues, it may lead to bowel stricture and intestinal obstruction. Intestinal obstruction in superior mesenteric vein thrombosis (SMVT) is a consequence of persistent ischemia, and its clinical manifestations appear rather later in disease evolution [1]. Most of the cases reporting intestinal obstruction as a result of SMVT had a two-stage presentation. First, there was a bowel ischemia, with patients describing abdominal pain, and second, they developed a bowel stricture and/or an intestinal obstruction [5].

The portal vein (PV) is an important major vein, a continuation of the superior mesenteric vein (SMV) that drains into the liver. Its name is based on its convergence with the splenic vein. The thrombosis of the PV (PVT) can be partial or complete. It frequently occurs secondary to primary or metastatic malignancies of the liver; cirrhosis from malignant and non-malignant diseases; inflammatory disorders, for example, Behçet syndrome; acute and chronic pancreatitis; abdominal infections; and hematological disorders including myeloproliferative or coagulation disorders. In the absence of malignancy and cirrhosis, systemic prothrombogenic conditions and local factors (acute pancreatitis, intra-abdominal infections, or abdominal trauma) are the main causes of PVT. Among these factors, myeloproliferative neoplasia appears as a major cause in 20–50% of the patients with PVT, and JAK2 V617F mutations were frequently found in PVT patients. Inherited and acquired thrombophilia appear to play a more important role in patients with Budd–Chiari syndrome than in those with PVT. Acquired viral infections or certain drugs may be risk factors for thrombosis [6]. For example, recent oral contraceptive use has long been acknowledged as a risk factor for thrombosis. Therefore, it is imperative for patients with recurrent thrombosis to have their past medical history and drug use history investigated.

According to previous data, the overall lifetime risk in the general population of developing PVT is estimated to be approximately 1%, as was concluded after a population study based on autopsy results [7]. Also, there are rare cases of non-cirrhotic non-malignant thrombosis of the PV as reported in the literature, with a lifelong prevalence in a general Western European population of about 0.3% [8]. The clinical manifestations of this condition are variable. Symptoms can be vague and include abdominal pain or fever. For some, perhaps most patients, thrombosis is found incidentally as there are cases of being asymptomatic. History taking or reviewing information in such cases may identify possible symptoms that may lead clinicians to suspect a PVT/MVT.

The key to early diagnosis is a high level of clinical suspicion. Severe abdominal pain out of proportion to physical examination findings should be assumed to be AMI until disproven.

The clinical scenario of a patient complaining of excruciating abdominal pain with an unrevealing abdominal examination is classic for early AMI. The reason for the pain being disproportionate to the clinical findings is that ischemia starts from the mucosa toward the serosa. That is why initially there is severe pain without clinical findings [9].

If the physical examination demonstrates signs of peritonitis, there is likely irreversible intestinal ischemia with bowel necrosis. In a study on AMI, 95% of patients presented with abdominal pain, 44% with nausea, 35% with vomiting, 35% with diarrhea, and 16% with blood per rectum. Approximately, one-third of patients present with the triad of abdominal pain, fever, and hemoccult-positive stool. Other patients, particularly those with delayed diagnosis, may present in extremis with septic shock [10].

Abdominal pain or detection of new-onset ascites is highly important and should prompt further assessment. In spite of that, it is quite common for patients to present with complications, for example, variceal bleeding from portal hypertension or spontaneous bacterial peritonitis, considering that PVT is commonly secondary to chronic liver diseases. PV is formed with SMV and splenic veins joining at the posterior side of the pancreas; it brings blood from the gastrointestinal tract to come up with 70% of the hepatic blood supply. Similar to any venous thrombosis at other sites, PVT and/or MVT can be caused by vessel endothelial injury, a low flow state, or the existence of thrombophilia [11]. According to a large population-based autopsy study, the basic etiology in about 22% to 28% of cases is cirrhosis. The prevalence rate of PVT in patients with cirrhosis depends on the severity of the liver disease (1% in compensated cirrhosis to 25% in patients who need a liver transplant) [12].

In a recent longitudinal multicenter randomized trial that included patients with Child A and B cirrhosis, the reported cumulative incidence rate of PVT was 4.6%, 8.2%, and 10.7% at 1, 3, and 5 years, respectively. PVT may be classified as acute or chronic, partial or complete, occlusive or non-occlusive. Additionally, among patients with cirrhosis, this could be caused by a bland or tumor thrombus [9].

PVT secondary to digestive malignancies has rarely been reported, with gastric cancer being one of the already published cases [13].

Regarding clinical features, abdominal pain represents the most common symptom of PVT or MVT. Physical examination is discordant most of the time, as it does not reveal significant modifications that compare to the severity of pain. There exist other nonspecific symptoms, for instance, vomiting, nausea, diarrhea, and fever, that can occur in 10% to 15% of cases and are caused by an underlying inflammatory intra-abdominal process, pylephlebitis with liver abscess, or intestinal infarction from MVT [14].

Among those suffering from cirrhosis, PVT can be asymptomatic, or patients may present with abdominal pain and/or with aggravation of pre-existing digestive symptoms or modification of liver function tests. It is mandatory to exclude tumor thrombus. Chronic PVT patients often present symptoms related to portal hypertension. Chronic PVT can be caused in approximately 1% of patients by biliary cholangiopathy because of peribiliary vascular collaterals compressing extrahepatic bile ducts, which manifests with cholangitis. Management of PVT in cirrhotic patients led to many discussions and is nowadays a controversial subject [15].

Clinical manifestations of MVT are variable. Symptoms usually encountered are abdominal pain, abdominal distension, eating-related symptoms, nausea, vomiting, hematemesis or hematochezia, and diarrhea. The presence of peritonitis may indicate a poor prognosis for patients with acute mesenteric ischemia. The regular course of MVT is insidious in comparison to the rapid fulminate course of mesenteric arterial occlusion. MVT can be categorized into acute, sub-acute, and chronic variants based on thrombus formation rapidity.

In a resting state, the bowel can tolerate an essential reduction in blood flow. According to existing data, only one-fifth of capillaries are needed in order to provide adequate oxygen delivery to tissues. However, in times of need, such as during stress status, oxygen extraction can be augmented by the intestinal mucosa. Sustained mesenteric ischemia exceeds the ability of capillaries to provide enough oxygen, triggering an inflammatory response, which eventually will lead to intestinal mucosal necrosis, a highly life-threatening complication. In an acute MVT, the occlusion is typically rapid and complete and leads to insufficient collateral development. In such cases, the evolution towards bowel ischemia and rarely perforation peritonitis should be foreseen [16,17].

In patients that enter the chronic phase, the development of collaterals will ensure venous drainage in order to avoid bowel ischemia but, in some cases, may not be sufficient to prevent bowel infarction and transmural necrosis [18].

Several reports have suggested the possibility of a two-stage evolution of the natural history of some cases of PVT; after initial treatment with a partial/complete resolution, there can be a second evolution with complications related to the first stage that require operative intervention. During the healing phase of intestinal ischemia, a subgroup of patients may develop fibrosis, which may progress and lead further to intestinal stricture and subsequent intestinal obstruction. There are also data reported in the literature revealing patients who, after an initial favorable response, developed avascular necrosis and tissue exfoliation of the intestinal mucosa in the following phase. If intestinal obstruction appears in evolution, it usually develops later on, or it may become symptomatic several weeks after an acute MVT. Evidence of this potential evolution has been reported in histopathological examination of previous case reports [19,20].

## 2. Diagnosis

Timing is very important because establishing the diagnosis is of tremendous importance, as it will lead to the correct management of the patient. A complete diagnosis aims to define if the thrombosis is acute or chronic, its extension, and its repercussions [15].

Virchow’s triad—hypercoagulability, endothelial injury, and reduced or impaired blood flow—is well known by all physicians, but in clinical practice, clear evidence may be difficult or sometimes impossible to prove. Literature data abound in information related to hypercoagulability states. Hypercoagulability may be found in non-cirrhotic and cirrhotic patients.

Frequently detected biochemical abnormalities are an increase in white blood cell (WBC) count and an increased C reactive protein (CRP) level, but these are nonspecific markers, as WBCs may be associated with an inflammatory systemic response. Additionally, a patient with a high lactate level, a high white cell count, and abdominal pain should raise suspicion of bowel ischemia, although these factors are not diagnostic.

One of the biochemical modifications that is frequently detected is the markedly increased CRP value. This frequent finding has been previously reported and has led to the prescription of antibiotics. However, as was shown for deep vein thrombosis, inflammation may be mediated by the release of cytokines, and it does not necessarily indicate infection.

Regarding other thromboses, patients with PVT and/or MVT are recommended a routine hematological and biochemical exam including D-dimers. For the diagnosis of PVT, a Doppler ultrasound examination is prescribed as the initial screening test. The advantages of ultrasound are well known; for example, it is inexpensive, free of radiation exposure, largely available as a bedside examination, and repeatable and has a high negative predictive value. However, the technique is operator- and patient-dependent, and the detection of a filling defect in the PV suggestive of PVT has a positive predictive value of only 86% to 97% [21]. Ultrasound, besides diagnosing thrombosis, may assist in interventional procedures such as guiding the transjugular puncture of the PV.

Contrast-enhanced computed tomography (CECT) or magnetic resonance (MR) scanning is usually recommended for confirming the diagnosis, identifying a tumor thrombus, and assessing the mesenteric vascularization (MVT, bowel ischemia, gangrene, and bowel perforation) before treatment is started. Literature data indicate that the presence of an intraluminal filling defect on a CECT or MR scan is quite accurate; its accuracy is up to 90% for the diagnosis of MVT and may increase to 100% with the use of the multidetector technique. Other important findings that transabdominal imagistic techniques can identify are portosystemic collaterals and cavernoma with serpiginous vessels replacing the PV, but these suggest previous PVT or a chronic diagnosis of PVT [22].

Here we illustrate the importance of CECT, which describes not only the characteristics of the thrombosis but also the consequence since there is ischemia of the distal ileal loops—see Figure 1.

As the clinical presentation of MVT is usually nonspecific, a CECT scan of the abdomen is often required for its diagnosis. CECT presents over 90% sensitivity for the diagnosis of MVT [23]. Along with venous thrombosis, CECT can also assess intestinal ischemia, which provides information for selecting eligible patients for a conservative approach. The detection of the presence of collaterals in a CECT scan of the abdomen helps in differentiating acute MVT from chronic MVT [15]. MR cholangiography coupled with MR portography is a non-invasive technique that allows the detection of biliary cholangiopathy in patients with chronic PVT [24].

Concerning any thrombosis, if there is no obvious etiology, patients without cirrhosis are recommended workup for the detection of thrombophilia. For patients with cirrhosis, if they have a family history of thrombosis, which also involves the hepatic veins, thrombophilia workup ought to be considered [25].

## 3. Management of MVT and PVT

The desired management aims in the matter of MVT and/or PVT are to prevent bowel infarction, perforation, peritonitis, and recurrence of the disease. Despite advances in treating thromboembolic disease, acute MVT can reach a 30-day mortality of up to 20–32% in severe cases.

When SMV thrombosis is established, interventional or surgical methods can be applied to perform the thrombectomy. The use of interventional therapy for this kind of vascular disease has increased recently. It has the potential to reduce the duration of hospitalization and trauma and hasten the postoperative recovery and the restoration of the intestinal tract’s infarcted blood vessels. The surgical procedures of interventional therapy involve fragmentation, crushing, and suction, especially percutaneous mechanical thrombectomy, which can lead to a reduction in the dose of thrombolytic agents and an acceleration of a patient’s postoperative recovery. The direct, local endovascular injection of the thrombolytic agent into the catheter proved to be successful in increasing the thrombolytic effect and diminishing associated bleeding problems. In patients with intestinal ischemia who received interventional therapy, the incidence of postoperative acute complications such as pulmonary or renal failure and death was significantly reduced compared with surgical treatment, which can remarkably improve their prognosis [26].

The treatment for MVT involves anticoagulation, supportive treatment, and surgery. Whilst emergency surgery is recommended in cases of bowel gangrene and perforation with peritonitis for acute MVT, for chronic MVT, surgery is required in patients who present intestinal stricture, severe gastrointestinal bleeding, small bowel perforation, and obstruction. Because of the development of severe metabolic acidosis caused by intestinal necrosis, fluid resuscitation and serial electrolyte monitoring are necessary.

Acute thrombosis may resolve or progress to cavernous transformation of the portal vein and portal hypertension within 2–4 weeks, which will require lifelong medical care. Knowing that interventional treatment in conjunction with anticoagulation has a higher response rate and a higher risk of serious adverse events (SAEs), it may preferably be used as a second-line option when medical treatment fails. However, as time proceeds, the interventional treatment’s response diminishes, making technical success less probable, SAEs recurrent, and stent implantation always imperative. This calls for a risk stratification model, authorizing an early interventional treatment when an unfavorable course of the medical treatment alone can be predicted [12].

In case of clinical status deterioration or if peritonitis was diagnosed in a patient with MVT, laparotomy is obligatory. During the exploratory laparotomy in the acute MVT, the macroscopic limits between the ischemic bowel and viable bowel are often unclear. One option is to resect the segments of the nonviable bowel, performing an enterectomy with anastomosis. Other options when enteral viability is not certain are to resect a longer segment of bowel and perform an intestinal anastomosis, close the abdomen, and perform a second-look procedure 24 to 36 h later; resect the bowel with irreversible ischemia, staple off the questionable ends of the bowel, and perform a second-look procedure 12 to 36 h later; or perform an enteral stoma. Unfortunately, a large enteral resection will cause short bowel syndrome postoperatively—see Figure 2.

In selected cases, open surgical thrombectomy may be an option [3,27]. When an emergency laparotomy must be performed, endovascular treatment with thrombolysis is contraindicated. If there are signs of extensive intestinal ischemia, which is rather rare, one possible strategy may be surgical thrombectomy combined with bowel resection of the gangrenous bowel, heparinization, and second-look laparotomy. In cases of frank transmural intestinal necrosis, uncertain bowel viability, or severe ischemic bowel lesions, bowel resection should be carried out. Nonetheless, peritonitis symptoms are not always indicative of transmural intestinal infarction, and conservative treatment is still an option for some patients who experience rebound tenderness. Primary bowel anastomosis after resection is recommended. This is a straightforward treatment in the typical patient in whom a short segment of the small bowel is affected—see Figure 2

The longer-term risks associated with short bowel syndrome make this conservative approach preferable to surgical resection, although this likewise raises the possibility of anastomotic leakage, when bowel ischemia does not appear to be present, particularly when methylprednisolone is used in conjunction with it [28].

Immediately after the diagnosis is made, systemic anticoagulation should be initiated as the first-line therapy for MVT. Anticoagulation is regarded as a fundamental treatment option and can effectively prevent the extension of the thrombosis or re-thrombosis. Further, it may lead to the recanalization of the occluded veins and improve intestinal reperfusion. This also lowers the risk of bowel infarction and allows time for venous collateral vessels to develop when the thrombus is not fully occlusive. Given that low-molecular-weight heparin has more advantages than unfractionated heparin, including a superior safety profile, ease of administration, and no need for routine laboratory monitoring, it is preferred [29]. Still, in some patients, such as those with a risk of bleeding, unfractionated heparin may be a safer option.

Recent data recommend that direct oral anticoagulants may represent a possible course of action in non-cirrhotic acute PVT. Treatment with non-vitamin-K-dependent oral anticoagulants such as drug-targeting factors Xa (Apixaban, Rivaroxaban) or thrombin (Dabigatran) could prevent recurrent thrombosis.

Testing for genetics can be done in order to exclude hereditary causes of MVT. To make an accurate diagnosis and offer optimal treatment, we can also investigate the genetic history of the family. We can also offer early preventive treatments to family members who might experience a thrombotic episode [25].

There are many anticoagulation agents available, and there are pros and cons of these. The proper agent for the induction regimen and, thereafter, the maintenance of the anticoagulation regimen should be taken into consideration. The standard of therapy for long-term management is anticoagulation with warfarin at a targeted international normalized ratio of 2.0 to 3.0. For some reversible etiologies, the majority of researchers advise continuing anticoagulant therapy for a minimum of six months following diagnosis to prevent thrombosis recurrence; nevertheless, certain reports continue to question the benefit. Patients with hereditary thrombophilia require lifelong anticoagulant medication in order to prevent further complications [30].

Anticoagulation treatment raises the survival rate of patients with acute MVT, reduces the risk of bowel ischemia or the need for surgical procedures, and increases recanalization rates. In one study, anticoagulation resulted in partial recanalization in 56% of patients and complete recanalization in 44% [31].

Choudhry et al. [32] studied a population group that had PVT secondary to inflammatory disease and reported that PV recanalization was higher and death rates were lower in those who were anticoagulated. Plessier et al. [33] revealed that early anticoagulation increased the recanalization and reduced bleeding risk. In-hospital mortality among identified and actively treated patients has decreased toward 20%, probably because of earlier detection with CT and the identification of a higher proportion of patients without peritonitis not requiring laparotomy. Anticoagulation with or without bowel resection in patients with acute MVT has also improved survival in comparison with observation alone. The overall 30-day survival in a contemporary series was 80% (41 of 51), and the estimated 5-year survival for the 51 patients was 70%.

In a randomized clinical trial, prophylactic anticoagulation reduced the incidence of PVT while improving liver function for patients with Child–Pugh class B or C cirrhosis. However, before implementing this as a standard procedure, larger multicenter studies are required [34].

Transcatheter thrombolytic injection therapy or transjugular intrahepatic portosystemic shunt (TIPS) are alternative approaches in cases with a contraindication for anticoagulation or an aggravated clinical situation, regardless of correct anticoagulation. These procedures are associated with high morbidity and an increased risk for bleeding complications [35].

Yang et al. presented a case in which a patient treated by transarterial thrombolysis via a catheter placed in SMA had no residual thrombus visible in the SMV, but after 4 weeks was readmitted with severe small bowel dilation for which surgery was mandatory. The free interval was variable, ranging from hours to months [36].

A prospective, multicenter study of acute non-cirrhotic, non-malignant PVT has found that there was a statistically significant difference in favor of interventional treatment (transjugular thrombus fragmentation, local thrombolysis with or without stent implantation) over medical treatment [12].

Intagliata et al. also questioned if there is a role for TIPS in the treatment of cirrhotic PVT and portal hypertension, and they concluded that recanalization is a challenging but safe and successful strategy as a bridging technique to transplantation [15,37].

Though anticoagulant therapy is successful in many cases, in which in first step does not demand surgery, as we can see, bowel stricture infarction, perforation, and peritonitis can develop, in spite of anticoagulant therapy, and in these cases, surgery is usually demanded [38].

Follow-up and monitoring for potential complications are also important. A chronic complication is the development of portal hypertension or the unusual occurrence of an acute intestinal infarction, which has a high mortality rate of up to 60%. The literature describes a case of intact necrotic intestinal mucosal tissue shedding and necrosis due to SMV thrombosis which occurred 5 months after an initial PVT and MVT [20].

Portal hypertension, which is clinically manifested by esophageal varices resulting from portosystemic shunting, frequently presents complications. A few less common side effects include the development of ascites, hypersplenism, and cavernous transformation of the portal vein. Previous research has inconsistent follow-up times and durations, and there are no established norms for reassessment. According to Hall et al., there were variations in the follow-up periods among studies, with some evaluating patients between zero and six years, others at twelve months or more, and some at less than twelve months. A patient’s surveillance interval must be adjusted, as there are some patients with serious side effects who may require more frequent follow-ups than others with a stable and asymptomatic thrombus development [39].

## 4. Discussion

The prevalence of AMI has changed in recent decades. The prevalence of acute mesenteric occlusion among patients with an acute abdomen may vary from 17.7% in emergency laparotomy to 31.0% in laparotomy for elderly non-trauma patients. Mesenteric arterial embolism has decreased in prevalence, accounting for 25% of cases. Mesenteric arterial thrombosis was the second most common cause of mesenteric ischemia; it historically accounted for 20–35% and recently increased in prevalence to 40% [3,40].

NOMI represents 25% of cases, and it is also increasing in prevalence compared to the historical cohort because of an increased number of critically ill patients and overall improvement of intensive care. Although the mechanism is still unknown, heart failure, renal failure, cardiac surgery using cardiopulmonary bypass, and the use of catecholamine are reported as risk factors [4].

The etiology of AMI has changed over the years with increasing percentages of acute arterial thrombosis due to atherosclerosis, which may in part be explained by modern anticoagulant therapy used for the treatment of atrial fibrillation.

The incidence of AMI has increased exponentially with age. In patients over 75 years of age, AMI is a more frequent cause of acute abdomen than appendicitis. The incidence of AMI in an 80-year-old is ten times higher than that in a 60-year-old patient [41]. Abdominal compartment syndrome with very high intra-abdominal pressure may cause bowel ischemia that is complicated by ischemia–reperfusion injury when decompression laparotomy is performed [42].

AMI has also been encountered in patients with COVID-19, probably in connection to large vessel thromboembolic events as well as to small vessel thrombosis caused by hypercoagulability and fibrinolysis shutdown [43].

The condition known as MVT, which causes 5–15% of cases of acute mesenteric ischemia, remains a life-threatening disease. Compared to the inferior mesenteric vein, the SMV develops thrombosis more frequently. This thrombosis may progress rapidly, despite it being characterized by an insidious onset and an early nonspecific clinical presentation. From an anatomical perspective, collateral vessels are absent in the SMV. A bowel infarction may result from a full obstruction followed by poor venous drainage. The position and extent of the thrombus, the size of the implicated arteries, and the depth of bowel-wall ischemia are the main factors influencing abdominal discomfort, which is the classic clinical indication of SMVT [44]. As peritonitis occurs, the symptoms in the abdomen gradually intensify and indicate a complete thrombosis of the SMV. Accurate and prompt diagnosis is critical to the effectiveness of acute SMVT treatment. A sensitive and specific laboratory marker for SMVT does not exist. Thrombosis can be identified by D-dimer testing, and the presence of a thrombus can cause the blood level to rise noticeably. However, D-dimer testing has not been thoroughly examined in the assessment of SMVT, and it can be elevated as a result of abdominal inflammation [45].

The most efficient method is a CECT scan with arterial and portal phases, which has a high sensitivity (96%) and specificity (94%). When intestinal ischemia is present, CECT images can identify extramural–nonvascular, vascular, and mural lesions, providing a prompt and precise diagnosis [46]. Based on its cause, MVT can be classified as primary or secondary. Primary hypercoagulable states or prothrombotic disorders, neoplasms, myeloproliferative disorder, diverse inflammatory conditions, portal hypertension, postoperative circumstances, trauma, or pregnancy can all lead to secondary MVT. Influenza and COVID-19 infections may result in SMVT in certain individuals [47].

MVT is referred to as primary or idiopathic when no etiologic or predisposing factors have been discovered. Usually, thrombophilia and malignancies should be ruled out during the patient’s examination [48,49]. Due to improved diagnostic performance and a greater understanding of predisposing variables, the percentage of primary MVT keeps declining. After ruling out any coagulation factor deficiency or other potential causes, we decide on idiopathic MVT as the diagnosis [50].

In 1974, the first case of post-renal transplant MVT was reported. The patient experienced an iliofemoral venous thrombosis one month following transplantation; given the hypercoagulable state, the thrombosis worsened eight months later, and the patient died. The reason why the patient experienced acute SMVT following the successful kidney transplant remains unknown. Ten years following a successful renal transplant, a second case of sudden abdominal pain along with acute PVT and MVT was also documented. After receiving anticoagulant therapy, the patient recovered, and systemic lupus was identified as the cause of the thrombosis.

The venous drainage of allografts occurs via the iliac veins, and the alterations in blood flow that follow may represent an unchangeable risk factor. It is unclear if immunosuppressive drugs raise renal transplant recipients’ risk of thrombosis. Tacrolimus may, in theory, cause thrombotic microangiopathy. Another contributing factor could be hypertension-induced endothelial damage, which is an essential factor for thrombosis formation [51].

The thrombophilic state is described in cases with nephrotic syndrome but not in chronic kidney disease. Even if the mechanism implicated remains suppositional, many prothrombotic effects might be involved [52].

PVT is a circumstance involving thrombotic occlusion of the PV. Either a partial or a complete occlusion is possible. PVT may be discovered in a variety of manners, with fevers and abdominal pain being the most common symptoms in symptomatic individuals or asymptomatic persons who are incidentally found while undergoing CECT. Patients with chronic PVT frequently experience portal hypertension, another common complication [53]. Usually, patients experience ascites, splenomegaly, and the risk of severe hemorrhage from rectal or esophageal varices. When PVT has no recognized cause, malignancies need to be considered [54]. Sometimes, there are no obvious factors that can be identified as the cause of the thrombotic event. Since idiopathic PVTs are uncommon, more research is necessary to determine the best course of treatment, including anticoagulation and follow-up [55].

## 5. Limitations of the Study

This literature review aimed to shed light on intestinal obstruction as the initial presentation of portal and mesenteric venous thrombosis. Unfortunately, there are no robust data, with data being based mainly on case reports or case series. Since this is not a frequent clinical encounter, there are limitations regarding the number of cases and the retrospective recollection of data.

## 6. Conclusions

An accurate diagnosis of portal and mesenteric venous thrombosis, early anticoagulant therapy, and timely surgical treatment are important for improving the patient survival rate, conserving as much of the bowel as possible, and avoiding and reducing postoperative recurrence and complications. Because multifactorial causes are frequent, the underlying cause should be investigated, and other causes should be excluded.

Though anticoagulant therapy is successful in many cases and as a first step does not demand surgery, as has been shown, bowel stricture infarction, perforation, and peritonitis can develop in spite of anticoagulant therapy, and in these cases, surgery is usually demanded and may be lifesaving.

## Figures and Tables

**Figure 1 diagnostics-14-00304-f001:**
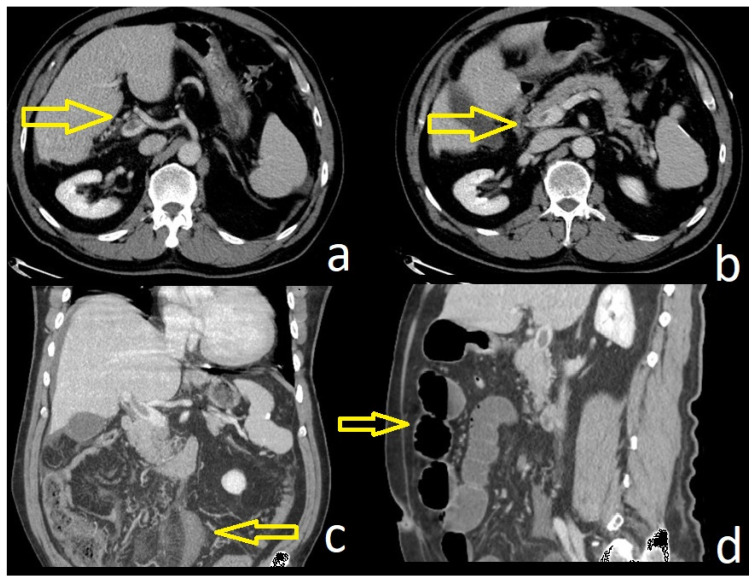
CECT aspects of a patient with concomitant PVT and MVT (**a**,**b**); ileal loops with edema (**c**) and ileus (**d**).

**Figure 2 diagnostics-14-00304-f002:**
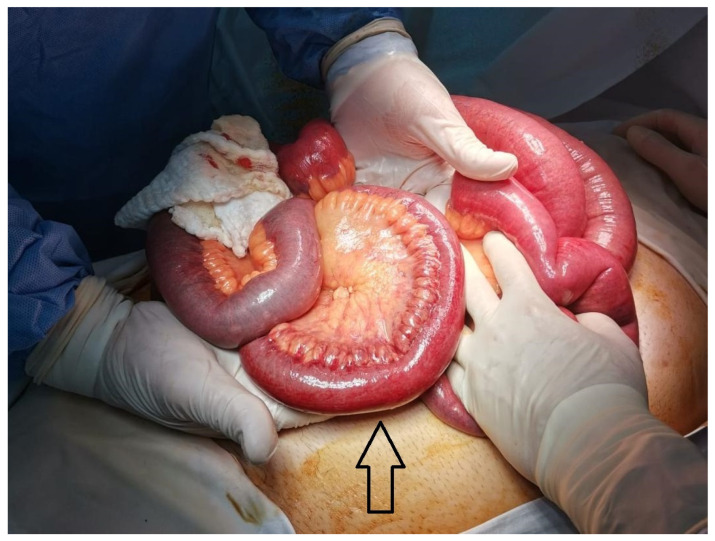
Intraoperative findings, macroscopic appearance of ischemic ileal loops due to concomitant PVT and MVT (marked with arrow), which required an enterectomy with anastomosis. The figures are original.

## Data Availability

Not applicable.

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
