# Peer review of "Intestinal Obstruction as Initial Presentation of Idiopathic Portal and Mesenteric Venous Thrombosis: Diagnosis, Management, and Literature Review"

_diagnostics, 2024, doi:10.3390/diagnostics14030304_

Round 1

Reviewer 1 Report

Comments and Suggestions for Authors

Comments to the authors

    This group from Romania first author Stancu R presents a review of a quite unusual complication of mesenteric thrombosis intestinal obstruction related to small bowel ischemia.  The benefit of this review is that many perhaps most clinicians will not think of mesenteric thrombosis as a cause of intestinal obstruction but a history of PVT or SMVT (if it was even diagnosed)  should then alert them to the potential etiology and pathogenesis.

Major comments

1. Introduction.  I suggest that you emphasize more the fact that intestinal obstruction is quite unusual for mesenteric ischemia

2. Page 2 line 59 hard for me to believe that PVT has a risk of 1% on the general population

3. Page 2 lines 72-79 a longitudinal study of what patient group? Cirrhosis? You need to define the group studied and you need a reference here which is lacking

4. Your 3rd, 4th, and 5th paragraph in the diagnosis section needs to be focused on saying that nonspecific tests such as WBC and CRP are of little help but the. Focus on the first suspicion of Thrombosis and then discuss hypercoagulopathy and hematologic tests of thrombosis - d dimers, etc. B t in patients with intestinal obstruction, a late effect of a prior e[isode of ischemia, do you expect to find an increase in these hematologic tests which are usually used in acute cases of thrombosis.  Also in those without  cirrhosis, a much more involved hematologic.-based diagnostic approach needs to be described

5.page 4 figure 1 you need to add arrows to point out the thrombus

6. Lines 153-154. This is true for mesenteric vein thrombosis but not for clot in the smaller mesenteric veins

7. I think you put too much emphasis on surgical thrombectomy or  interventional thrombectomy/lysis

8. Figure 2 you mention intestinal obstruction  this does not look like a true mechanical obstruction but rather a limited ileus

9 page 6 line 225.  Here is where the majority of focus should be. i.e. on recognition and then aggressive anti coagulation to prevent extension of the thrombosis

10. Lines 239 - 257. Excellent.  Here is where the focus should be line 3-8 don’t you mean thrombosis and not embolization

Minor comments

1.abstract line 20.  I would add the terms “ gastroesophageal and  even small intestinal “ before the term “varies”

2. Abstract line 22 add the word “but” before the second  word “Thrombosis”

3. Abstract line  24.  Intestinal obstruction from mesenteric venous thrombosis is  rare  and I think you need to emphasize this.

4. Page 2 line 48 add here acute and chronic pancreatitis

5 page 2 line 55. What do you mean by drug factors, I am having problems understanding g this phrase

6. Page 2 lines 85. Pylephebitis is extremely unusual so I wouldn’t focus on it. The distant literature described this often but in current practice, Pylephlebitis is extremely rare - other causes  far outweighs pylephlebitis

7 page  2 lines 88-89. I do not agree with this statement. You are referring to intraheptic venous occlusion  but malignant tumor thrombosis of the PV is not a common. similarly your discussion of PVT related cholangiopathy is even more rare this should be reserved for a one sentence discussion of the rare forms of presentation of SMVT and PVT and limit the discussion of these once in a career presentations of mesenteric venous thrombosis.

8. Page 3 line 111 what do you mean by transluminal infarction. ? Don’t you mean just intestinal infarction or transmural necrosis?

PVT

9. Page 3 lines 131 -142.  WBC and CRP are is  too nonspecific and you should state that

10. Page 3 lines 146-147. Delete this I am not certain where you are going with  this part of the discussion

11. Page 4 line 153. Is Kodali a reference ?

12. Lines 149 and 150 why the focus on tumor thrombosis this is quite rare

13. Again 167-169. This cholangiopathy is so rare that it doesn’t require much  discussion

14. Line 237 DOAC and NOAC ARE  not a common abbreviations delete THEM

 I believe strongly that scientific reviews should be signed and request the editor to convey my name to the authors.  

Comments on the Quality of English Language

Author Response

Thank you for the suggestions and remarks that we have received. We strongly believe that your work, led us to improve our manuscript.

We have merged changes you suggested, together with those proposed by other reviewers.

We have also modified title, as it was suggested by another reviewer.

This group from Romania first author Stancu R presents a review of a quite unusual complication of mesenteric thrombosis intestinal obstruction related to small bowel ischemia.  The benefit of this review is that many perhaps most clinicians will not think of mesenteric thrombosis as a cause of intestinal obstruction but a history of PVT or SMVT (if it was even diagnosed) should then alert them to the potential etiology and pathogenesis.

 Major comments

  1. Introduction.  I suggest that you emphasize more the fact that intestinal obstruction is quite unusual for mesenteric ischemia

We have underlined the main idea

  1. Page 2 line 59 hard for me to believe that PVT has a risk of 1% on the general population – In the original paper, they have reported a risk of 1.1% as result of autopsy studies.

We have inserted also the reference which was missing.

  1. Page 2 lines 72-79 a longitudinal study of what patient group? Cirrhosis? You need to define the group studied and you need a reference here which is lacking

We have inserted information and also the reference which was missing

  1. Your 3rd, 4th, and 5th paragraph in the diagnosis section needs to be focused on saying that nonspecific tests such as WBC and CRP are of little help but the. Focus on the first suspicion of Thrombosis and then discuss hypercoagulopathy and hematologic tests of thrombosis - d dimers, etc. B t in patients with intestinal obstruction, a late effect of a prior episode of ischemia, do you expect to find an increase in these hematologic tests which are usually used in acute cases of thrombosis.  Also, in those without cirrhosis, a much more involved hematologic. -based diagnostic approach needs to be described

We have inserted information

  1. page 4 figure 1 you need to add arrows to point out the thrombus

We have marked with arrows on the image.

  1. Lines 153-154. This is true for mesenteric vein thrombosis but not for clot in the smaller mesenteric veins

We emphasized

  1. I think you put too much emphasis on surgical thrombectomy or interventional thrombectomy/lysis

With the modifications performed, it is not so much emphasized as before, in our opinion, and we described more about surgical treatment.

  1. Figure 2 you mention intestinal obstruction this does not look like a true mechanical obstruction but rather a limited ileus

We agree, it wasn’t a mechanical obstruction, but it was a limited ileus caused by ischemia because of PVT an MVT. We performed in this case an enterectomy with anastomosis.

9-page 6 line 225.  Here is where the majority of focus should be. i.e. on recognition and then aggressive anti coagulation to prevent extension of the thrombosis

We emphasized

  1. Lines 239 - 257. Excellent.  Here is where the focus should be line 3-8 don’t you mean thrombosis and not embolization

 We emphasized

Minor comments

1.abstract line 20.  I would add the terms “gastroesophageal and even small intestinal “before the term “varies”

We inserted those terms.

  1. Abstract line 22 add the word “but” before the second word “Thrombosis”

We inserted that word.

  1. Abstract line 24.  Intestinal obstruction from mesenteric venous thrombosis is rare and I think you need to emphasize this.

 We emphasized

  1. Page 2 line 48 add here acute and chronic pancreatitis

 We added those words

  1. page 2 line 55. What do you mean by drug factors, I am having problems understanding g this phrase

 We have inserted information and also a reference

  1. Page 2 lines 85. Pylephebitis is extremely unusual so I wouldn’t focus on it. The distant literature described this often but in current practice, Pylephlebitis is extremely rare - other causes far outweighs pylephlebitis

We totally agree, but if we would have left it unmentioned, we would have received comments from other reviewers. It is just one sentence; therefore, we consider it should be maintained.

7 page 2 lines 88-89. I do not agree with this statement. You are referring to intrahepatic venous occlusion but malignant tumor thrombosis of the PV is not a common. similarly, your discussion of PVT related cholangiopathy is even more rare this should be reserved for a one sentence discussion of the rare forms of presentation of SMVT and PVT and limit the discussion of these once in a career presentation of mesenteric venous thrombosis.

We agree about that, we mentioned that this represents 1% of cases.

  1. Page 3 line 111 what do you mean by transluminal infarction? Don’t you mean just intestinal infarction or transmural necrosis?

We agree, we modified.

 In patients that entered in the chronic phase, development of collaterals will ensure venous drainage in order avoid bowel ischemia but, in some cases may not be sufficient to prevent bowel infarction and transmural necrosis

  1. Page 3 lines 131 -142.  WBC and CRP are is too nonspecific and you should state that

We have inserted this information also we have rephrased all the paragraph

  1. Page 3 lines 146-147. Delete this I am not certain where you are going with  this part of the discussion.

We have rephrased all the paragraph

  1. Page 4 line 153. Is Kodali a reference? Yes, it was a reference we have inserted it
  2. Lines 149 and 150 why the focus on tumor thrombosis this is quite rare

We agree

  1. Again 167-169. This cholangiopathy is so rare that it doesn’t require much  discussion

We agree

  1. Line 237 DOAC and NOAC are not common abbreviations delete them

We have deleted them. Indeed, they appeared just once!

Reviewer 2 Report

Comments and Suggestions for Authors

Thrombosis of the portal venous system may occur in a variety of clinical conditions: the most common is hepatocellular carcinoma (1). Moreover intestinal ischemia commonly occurs after arterial thrombosis or embolism.

1.   In this review, authors investigate intestinal obstruction as initial presentation of idiopathic portal and mesenteric venous thrombosis. I suggest to ameliorate historical bakground.

2.   In literature, there are evidence of intestinal obstruction as initial presentation of idiopathic portal and mesenteric venous thrombosis. Thrombosis is often idiopathic in nature, with up to 49% having no identifiable cause. I suggest to investigate risk factors: abdominal inflammation and systemic thrombophilias (2).

3.   Statistical examination is poor. They then don’t measure the potential exposure of interest.

4.   The last decade has witnessed increased recognition of the value of literature reviews; What kind of review is this? The investigator roughly defines diagnosis and management for advancing understanding and decision making.

5.   The conclusions are not consistent with the evidence and arguments presented and they not address the main question posed.

6.   The references are inappropriate (1-2).

7.   Figures are appropriate but table are inappropriate: I suggest to include a “typology source diagnosis” and a “typology source management” (3).

References

1.   Vannelli A, Fiore F, Del Conte C, Rivolta U, Corsi C. Pylethrombosis associated with gastric cancer in Moschcowitz's disease: successful management with anticoagulant. Report of a case. Tumori. 2004 Mar-Apr;90(2):259-61. doi: 10.1177/030089160409000220. PMID: 15237595.

2.   Dan Nicolae Bele. Complicated idiopathic portal and mesenteric venous thrombosis: A case report. Acta Marisiensis - Seria Medica 2023;69(2):138-140 doi: 10.2478/amma-2023-0020

Author Response

We kindly thank you for all your suggestions that contributed to improve our manuscript.

We have merged changes you suggested, together with those proposed by other reviewers.

We have also modified title, as it was suggested by another reviewer.

Thrombosis of the portal venous system may occur in a variety of clinical conditions: the most common is hepatocellular carcinoma (1). Moreover, intestinal ischemia commonly occurs after arterial thrombosis or embolism.

  1. In this review, authors investigate intestinal obstruction as initial presentation of idiopathic portal and mesenteric venous thrombosis. I suggest to ameliorate historical background.

We have added more information and rewritten some of the paragraphs.

  1. In literature, there are evidence of intestinal obstruction as initial presentation of idiopathic portal and mesenteric venous thrombosis. Thrombosis is often idiopathic in nature, with up to 49% having no identifiable cause. I suggest to investigate risk factors: abdominal inflammation and systemic thrombophilias (2).

We have added more information and rewritten some of the paragraphs

  1. Statistical examination is poor. They then don’t measure the potential exposure of interest.

We wrote about the limitation of the study at the end of the paper, there are only case reports and case series

  1. The last decade has witnessed increased recognition of the value of literature reviews; What kind of review is this? The investigator roughly defines diagnosis and management for advancing understanding and decision making.

We have modified the title, as it was suggested by another reviewer without terms diagnosis and management.

  1. The conclusions are not consistent with the evidence and arguments presented and they not address the main question posed.

We have revised the conclusions.

  1. The references are inappropriate (1-2).

We added more references

  1. Figures are appropriate but table are inappropriate: I suggest to include a “typology source diagnosis” and a “typology source management” (3).

We pointed on the figures with arrows.

References

  1. Vannelli A, Fiore F, Del Conte C, Rivolta U, Corsi C. Pylethrombosis associated with gastric cancer in Moschcowitz's disease: successful management with anticoagulant. Report of a case. Tumori. 2004 Mar-Apr;90(2):259-61. doi: 10.1177/030089160409000220. PMID: 15237595.
  2. Dan Nicolae Bele. Complicated idiopathic portal and mesenteric venous thrombosis: A case report. Acta Marisiensis - Seria Medica 2023;69(2):138-140 doi: 10.2478/amma-2023-0020

We have inserted these references also.

Reviewer 3 Report

Comments and Suggestions for Authors

Dear Authors, 

Mesenteric venous thrombosis is an important issue that requires investigation and special attention, so that any study that brings new information deserves to be published. In any case, your manuscript requires extensive revision before it can be published.

Title: If you mentioned in the title that it is a literature review, I think the words "diagnosis and management" are no longer necessary.

Introduction: please specify the aim of the study. 

I think that figure 1 should be sub-numbered and arrows should be added to clearly indicate the imaging aspects mentioned in the legend.

Are figure 1 and 2 original? If so, please specify. 

Line 153 - please revise the reference in brackets. 

The Management section must be divided into sub-chapters that deal separately with medical and surgical management. 

Line 215-225 - References are needed for these affirmations. 

I think a subsection with the limitations of the study is necessary. 

The Conclusions chapter needs to be redone to be more specific.

Author Response

We kindly thank you for all your suggestions that contributed to improve our manuscript.

We have merged changes you suggested, together with those proposed by other reviewers.

We have also modified title, as it was suggested.

Mesenteric venous thrombosis is an important issue that requires investigation and special attention, so that any study that brings new information deserves to be published. In any case, your manuscript requires extensive revision before it can be published.

Title: If you mentioned in the title that it is a literature review, I think the words "diagnosis and management" are no longer necessary.

We also consider that they were not necessary, therefore we have deleted them.

Introduction: please specify the aim of the study.

We have inserted this information

I think that figure 1 should be sub-numbered and arrows should be added to clearly indicate the imaging aspects mentioned in the legend.

Are figure 1 and 2 original? If so, please specify

Yes, they are. We have inserted also this information

Line 153 - please revise the reference in brackets. 

We have inserted it accordingly

The Management section must be divided into sub-chapters that deal separately with medical and surgical management. 

Line 215-225 - References are needed for these affirmations. 

We have inserted new references

I think a subsection with the limitations of the study is necessary. 

We have inserted a subsection with the limitations of the study.

The Conclusions chapter needs to be redone to be more specific.

We have revised the conclusions.

Round 2

Reviewer 1 Report

Comments and Suggestions for Authors

Comments to the authors 

Major comments  NONE 

1. Authors nice job on this review. Virtually all of my comments are to make it easier to read AND to make your English look even better than it does!   So please take all the following suggestions constructively these changes WILL make the editor’s job easier!

Minor

1. Page 1 line  25 and 26. I would reword this as “Intestinal obstruction  or more commonly ileus due to mesenteric venous thrombosis is rare and only a small number of cases have been reported”  I really think that this will further support your paper’s title

I would also suggest that you change the  word “obstruction” on line 27 to “obstruction/ileus” and also add intestinal ileus to the key words. This is because figure 1 shows an ileus rather than intestinal obstruction. I know that you want to emphasize obstruction,but ileus is more common, and a true obstruction would require a mechanical cause like a structure or severe serosal adhesions or a perforation that blocks the bowel

2. Page 2 line 44 change “branch” to “major vein” 

3. Page 2 line 59 and 60. Much better well done i would guess that many perhaps most of the 1% were asymptomatic i might even suggest that you highlight that. PVT clinically is quite rare and certainly does not cause symptoms in 1% of the population over their lifetime and on line 65 consider adding the phrase” perhaps most” after the word “some”

4. Page 2 line 74. Change Los blood flow to “a low flow state”

5. Pane page 3 lines 115-117. This is hard to read suggest you change it to several reports have suggested the possibility of a 2 a-stage evolution of the natural history of some cases of PCT; after initial treatment with a partial/complete resolution, there can be a 2nd evolution with complications related to the first a stage that the require operative intervention”.  This has the same thing but makes more sense.

6. Page 5 line 126 replace “precise” with “ define”

7. Page 5 line 154 change “imagistic technique may” to “trans abdominal imaging techniques can”

8. Page 5 lines 178 replace "the“  with “a” line 179 delete the word “significantly” that word implies statistical evaluation 

9. Page 6 line 183   after the phrase ”success of” change to  “direct, local endovascular”

10. Page 6 line 195 change to “cavernous transformation of the portal vein”

11. Page 6 lines 206-208 I would reword this as “Other options when enteral viability is not certain are to resect a longer segment of bowel and perform an intestinal anastomosis,  close and perform a second look procedure 24 to 36 hours later, or to resect the bowel  with irreversible ischemia and staple off the questionable ends of the bowel and perform a second look procedure 12 to 36 hours later, or to perform an enteral stoma——“

12. Page 6 line 220. End the second to last sentence with “ when bowel ischemia does not appear to be present”

13 Figure 2  when I look at the bowel to which the arrow points, I am not convinced that it had irreversible ischemia. Did you consider a second look procedure? 

14. Page 8 line 234 replace alternative with “etiology” 

15. Page 8 line 267 change “in defiance of  correct” to  “in which anti coagulation would be contraindicated”

16. Page 9 line 278 I believe the abbreviation is TIPS not just TPS  and does ref 17 really address TIPS? It looks like an isolated case report and if so, I would delete it I see that it is used online 288 appropriately 

17. Page 9 line 291 again.    Change to “cavernous transformation of the portal vein”

18. Line 307 I think you mean "thrombosis" rather than "embolization"

19. Line 315 what do you mean by “mural” symptoms

20 line 317 why not start the paragraph with “organ Transplantation:” and combine the next 2 paragraphs into one paragraph

21. Line 298 shouldn’t this be labeled “SUMMARY” rather than discussions- much or most of what follows from line 299- 356 has been said above

22. Lines 352-356. This seems out of place the splenic are the same as the mesenteric and portal veins. I would delete this altogether or add it up in the text above 

 I believe strongly that scientific reviews should be signed and request the editor to convey my name to the  authors Michael G Sarr  MD

Comments on the Quality of English Language

This submission is from a non native speaking set of authors  It is really really good but there are places that will require only minor changes I made quite a few to the authors but there are other places where a few phrases or words are not ideal for your journal so yes it will take about 15 minute for a copy editor to spruce it up all changes are minor

Author Response

Dear Reviewer, Thank you very much for your constructive comments and suggestions. 

For the Minor comments we modified in the text as follows:

  1. Page 1 line  25 and 26 - We agree, we changed it.  and also on line 27 to  and also added intestinal ileus to the key words. l
  1. Page 2 line 44   - We agree and we changed it.
  2. Page 2 line 59 and 60 -  We agree and we changed it and also on line 65 
  3. Page 2 line 74. We changed it
  4. Page 3 lines 115-117.  We agree and we changed it..
  5. Page 5 line 126 - We changed it
  6. Page 5 line 154  - We agree and we changed it.
  7. Page 5 lines 178 - We agree and we changed it. 
  8. Page 6 line 183 - We changed it
  9. Page 6 line 195 - We changed it
  10. Page 6 lines 206-208 - We agree with you and we replaced it.
  11. Page 6 line 220. We changed it.

13 Figure 2  when I look at the bowel to which the arrow points, I am not convinced that it had irreversible ischemia. Did you consider a second look procedure?

Yes, in that case we performed a second look procedure. We agree with you.

  1. Page 8 line 234 - we replaced alternative with "possible course of action" because we tried to say that direct oral anticoagulants are a treatment option.
  2. Page 8 line 267 we changed “in defiance of  correct” to  “regardless", there is another term contraindication in the same phrase.
  3. Page 9 line 278 - We changed it with TIPS of course. We deleted ref 17 from here. 
  4. Page 9 line 291 - We changed it 
  5. Line 307  We changed it to "thrombosis".
  6. Line 315 - We changed  to "lesions"

20 line 317 - We agree and we changed it.

  1. Line 298 shouldn’t this be labeled “SUMMARY” rather than discussions- much or most of what follows from line 299- 356 has been said above. You are right but the Journal has some patterns which we must respect. We maintained the word Discussions.
  2. Lines 352-356. - We deleted this phrase 

Reviewer 2 Report

Comments and Suggestions for Authors

In my opinion the manuscript has been sufficiently improved to warrant publication in Diagnostics

Author Response

Dear Reviewer, Thank you very much for your constructive comments and suggestions. 

Reviewer 3 Report

Comments and Suggestions for Authors

Dear authors,

Thank you for  answering almost all my questions or suggestions. I believe that the manuscript has been improved and can be published after a revision of some minor typographical errors.

Author Response

(The authors gave the same response as above.)
